# Longitudinal Analysis of Work-to-Family Conflict and Self-Reported General Health among Working Parents in Germany

**DOI:** 10.3390/ijerph17113966

**Published:** 2020-06-03

**Authors:** Lea-Sophie Borgmann, Petra Rattay, Thomas Lampert

**Affiliations:** Department of Health Monitoring and Reporting, Robert Koch Institute, Nordufer 20, Berlin 13353, Germany; rattayp@rki.de (P.R.); lampertt@rki.de (T.L.)

**Keywords:** work-to-family conflicts, self-reported general health, education, pairfam, longitudinal analysis, logistic regression, moderator analysis, predictive margins, Germany

## Abstract

The combination of work and family roles can lead to work-to-family conflict (WTFC), which may have consequences for the parents’ health. We examined the association between WTFC and self-reported general health among working parents in Germany over time. Data were drawn from wave 6 (2013) and wave 8 (2015) of the German family and relationship panel. It included working persons living together with at least one child in the household (791 mothers and 723 fathers). Using logistic regressions, we estimated the longitudinal effects of WTFC in wave 6 and 8 on self-reported general health in wave 8. Moderating effects of education were also considered. The odds ratio for poor self-reported general health for mothers who developed WTFC in wave 8 compared to mothers who never reported conflicts was 2.4 (95% CI: 1.54–3.68). For fathers with newly emerged WTFC in wave 8, the odds ratio was 1.8 (95% CI: 1.03–3.04). Interactions of WTFC with low education showed no significant effects on self-reported general health, although tendencies show that fathers with lower education are more affected. It remains to be discussed how health-related consequences of WTFC can be reduced e.g., through workplace interventions and reconciliation policies.

## 1. Introduction

Digitalization and globalization of the job market are leading to changes in job roles and new demands on employment (1). At the same time, workers are increasingly faced with challenges of organizing care for their children and elderly relatives as societies age [1,2]. Particularly for parents an increased flexibility at the workplace and extended availability of institutional childcare provide opportunities to reconcile work and family roles. In Germany, for example, the reconciliation of work and family lives is supported by governmental structures, e.g., with a parental leave scheme for mothers and fathers that covers up to twelve months of paid parental leave for one parent, which can be extended by two months of parental leave for the partner [3]. Since 2013, children from one year of age are also legally entitled to institutional childcare [4].

Besides these positive developments, new conflicts arise as the boundaries between work and family become blurred and the demands of both areas of life overlap. Theoretically, such conflicts are anchored in role theory. They are defined as inter-role conflicts in which demands from both work and family roles are not or only partially compatible. The resulting conflicts can act in two directions: ‘Work-to-family’ conflicts (WTFCs) arise when demands at work disrupt family life. ‘Family-to-work’ conflicts (FTWC) occur when demands in the family reach into the work sphere. Both directions of conflict are mutually dependent [5]. In the following introductory sections, we will refer to work-family conflict (WFC) as the general variable of exposure when describing findings from previous work.

Conflicts while reconciling work and family roles present themselves differently in the everyday lives of mothers and fathers: Although, in 2019, 77.5% of mothers in Germany between the ages of 15 and 64 were employed, more than two-thirds of them worked part-time only. For fathers, the employment rate was 85.7% in 2019, but less than one in ten of the actively employed fathers were part-time workers [6,7]. Similar to the employment rate, gender differences in paid parental leave also exist: In 2019, mothers planned to take an average of 11.7 months of paid parental leave, while fathers only intended to take 2.9 months on average [8]. Moreover, in 2019, only 34.3% of children younger than three years of age were in institutional care [9]. In summary, the political action to improve the reconciliation of work and family lives in Germany is limited and used to varying degrees by mothers and fathers. This might be one reason why about one third of parents in Germany report that their work frequently or permanently affects family life [3]. Additionally, nearly half of the employed women in Germany state that they were very often or often too exhausted after work to be able to take care of private or family affairs; while this affected only one-third of men [10]. In addition to these differences by gender, there are also differences within the groups of mothers and fathers [11]. For example, the maternal employment rate is particularly dependent on the education of mothers [12]. Accordingly, it can be assumed that WFC do not only occur among working mothers and fathers to varying degrees, but may also be influenced by social determinants such as education or household income.

### 1.1. Work–Family Conflicts and Health

Stress process theory assumes that stress is a causal antecedent of individual health and well-being [13,14]. It proclaims that stressors can result from various sources. One type of stressor stems from the reconciliation of work and family settings and is referred to as WFC. Empirical reviews have already shown that WTFC and FTWC are associated with general, mental, and physical health [15,16,17,18].

Studies on the longitudinal association of WFC and health are, however, rather scarce. The few existing publications show that WFC leads to poorer self-reported general and mental health over time [19,20,21,22,23,24]. However, none of the existing studies looked at the level of reported WFC across different points in time, leaving out the potential to examine cumulative effects. From stress process theory, it is assumed that experiencing WTFC at more points in time leads to worse health-related outcomes, compared to experiencing WTFC never or less often.

In addition, the majority of the available evidence is based on data from the U.S. and Canada. Since the political and cultural backgrounds play an important role for understanding the association of WFC and health these results can only be transferred to the German context to a limited extent [15]. Here, we contribute to the existing research by using longitudinal data from Germany and, thus, adding to the few studies available [25,26,27]. 

### 1.2. The Role of Social Determinants

According to stress process theory, larger structures of society shape how stress is experienced individually [14]. As the same stressor can have a different impact depending upon characteristics such as gender, mothers and fathers may perceive detrimental effects of WFC on their health to varying degrees since work and family roles are highly shaped by gender norms [28]. Prior research partly supports this association and shows that health burden caused by WFC differs between mothers and fathers, whereas mothers report stronger effects of WFC on e.g., physical health than fathers [16]. Other studies show no differences according to gender [29]. However, most of the other publications on WFC and health do not differentiate the results by gender [17,30,31]. Hence, the existing evidence does not permit a clear conclusion concerning differences between mothers and fathers with regard to the health effects of WFC [32]. 

One cause of the inconclusive results on gender differences can be a lack of differentiation within the groups of working mothers and fathers: reviews postulate that insufficient data is available to investigate the health effects of WFC, for example among single-parent families or parents living at risk of poverty [17,31]. In addition, only few other social determinants of health have been taken into account: One study showed that people with lower formal education were more affected by the health effects of WFC [19]. Another study from Japan presented that the association between WFC and self-reported general health was more evident among women with low household income, compared to women from higher-income groups. This interaction was not observed among men [32]. That differentiation in health research beyond pure stratification by gender categories can be of analytical importance has already been argued in various places [33,34]. Thus, we extend prior research as we present a differentiated view of the gender groups by adding the level of education to the longitudinal analyses.

### 1.3. Contribution and Research Questions

We contribute to the existing literature by combining population-level cohort data from Germany with a longitudinal analytical approach. Thus, we add to the limited number of longitudinal studies from Europe and Germany. We also present the first analysis looking at cumulative effects of WTFC on health. Furthermore, and also novel to the field, we employ an intersectional approach to identify subgroups among mothers and fathers by an intersection of gender and education. This allows us to further differentiate the subgroups of working mothers and fathers and thus contribute to disentangling the inconclusive evidence on gender differences presented in prior research. The following questions guided our research: 

Research question 1: To what extent are WTFC and self-reported general health associated? Are differences between working mothers and fathers observed?

Research question 2: Does WTFC have an effect on self-reported general health over time? Is this different for working mothers and fathers?

Research question 3: To what extent is the association of WTFC and self-reported general health moderated by education? Is moderation different for working mothers and fathers?

## 2. Materials and Methods

### 2.1. Data

Data were drawn from the relationship and family panel “pairfam” (Panel Analysis of Intimate Relationships and Family Dynamics), which collects data on partnership and family dynamics in Germany [35]. The longitudinal survey started in 2008 with randomly selected persons from the birth cohorts 1971–1973, 1981–1983, and 1991–1993. From these cohorts, stratified random samples of approximately 4000 interviews each were generated in wave 1 (total *n* = 12,402). The population consisted of all German-speaking persons living in private households in Germany. Personal interviews were conducted using the CAPI (computer assisted personal interview) method and lasted about one hour on average. The study participants have been surveyed yearly since 2008. So far, data from wave 1 (2008/2009) to wave 9 (2017/2018) have been published. In addition to the interviewees, who are referred to as “anchor persons”, their partners, parents, and children aged nine and over were also interviewed in each wave. Pairfam was approved by the ethics committee of the Faculty of Management, Economics, and Social Sciences of the University of Cologne. Further technical details on the collection of pairfam data are reported elsewhere [36]. 

For this study, all persons who completed wave 6 (T0) and wave 8 (T1) of the survey were included (total *n* = 6708), as only waves 6 and 8 included data on WTFC. To limit the analysis to working parents, all persons who stated that they were inactive, in education, or without children in the household at either time were excluded (*n* = 5361). In addition, those persons were excluded for whom no data on self-reported general health, on WTFC, and on the selected control variables were available (*n* = 189). The data for the remaining 1514 persons were checked for completeness, and no further exclusions had to be made.

### 2.2. Dependent Variable

Health was operationalized with self-reported general health at T1. This general health indicator has been proven to be a sufficient proxy measure for general, physical, and mental health [37]. All participants were asked the following question: “How would you describe your health status in the past four weeks in general?” The response was “bad”, “not so good”, “satisfactory”, “good”, and “very good”. The answers were grouped into “very good/good” and “satisfying/not so good/bad”. Dichotomization was implemented to provide results that are comparable to prior research and easy to interpret [20,32,38]. Categorizing the ordinal scale into a binary response does not have an impact on the estimated effects of covariates [39].

### 2.3. Independent Variable

In the pairfam dataset, a total of eight items on WFC were collected, four of which referred to WTFC and four to FTWC, respectively [40]. The scale was adapted from Carlson and Grzywacz [41] and translated into German by Wolff and Höge [42]. The focus of the present analysis lies on the four items measuring WTFC. For FTWC not enough variation was observed in the respective waves, as only about 10 percent of men and women reported high FTWC across waves 6 and 8. Following the question “Now we would like to know how your personal life and your work influence one another. To what extent do the following statements apply to you?” answers were collected for the following four items: (a) Because of my workload in my job, my personal life suffers. (b) Even if I do something with friends, partners, or family, I must often think about work. (c) After the stress of work, I find it difficult to relax at home and/or to enjoy my free time with others. (d) My work prevents me from doing things with my friends, partner, and family more than I’d like. The response options were on a Likert scale from “not at all” (1) to “absolutely” (5). A sum index was formed, with a minimum of 4 and a maximum of 20 points. Cronbach’s alpha of this scale was 0.80 (wave 6) and 0.77 (wave 8) for mothers and 0.74 (wave 6) and 0.75 (wave 8) for fathers, indicating good internal reliability. The index was dichotomized with 11 and more points being interpreted as “high WTFC” and 4 to 10 points as “low WTFC”. The cut point guarantees that participants have scored a 3 on the Likert scale in at least three items.

To allow for longitudinal analyses, participants were divided into four groups based on the prevalence of WTFC at T0 and T1: Persons with low WTFC at both times (T0: low WTFC, T1: low WTFC, reference group), persons with high WTFC at both times (T0: high WTFC, T1: high WTFC), persons with high WTFC at T1 only (T0: low WTFC, T1: high WTFC), and persons with high WTFC at T0 only (T0: high WTFC, T1: low WTFC). From a theoretical point of view, the groupings assume that people with high WTFC at both T0 and T1 experienced increased exposure compared to persons who reported high WTFC only once or never. In addition, the time gap between exposure and outcome was taken into account by differentiating between exposure to high WTFC at T0 and at T1 only. Thus, the division of participants into these groups allowed for analysis not only of longitudinal effects of WTFC on health over time but also for cumulative effects of WTFC on health at T1.

### 2.4. Control Variables

Models were controlled for age at T0 and area of residence at T1. The area of residence was included dichotomously as the residential region in Germany with the values “West” and “East” (area of the former German Democratic Republic). Models 1 to 5 are also controlled for self-reported general health at T1. Education (International Standard Classification of Education, ISCED) and household income were integrated into the models as sociodemographic control variables. Education was coded dichotomously as “low and medium education” and “high education”, with “high education” covering all tertiary educational qualifications. To categorize household net income, respondents were divided into three income groups of equal size. Dichotomization was achieved by combining the two low- and middle-income groups and contrasting them with the third of participants with the highest income.

Family-related variables taken into account were the number of children under 18 years of age in the household (“1”, “2”, and “more than 2 children”), age of the youngest child in three age groups (0–4, 5–13, and 14–17 years of age) and marital status with the values “never married”, “married/registered civil partnership” and “divorced”. Furthermore, information on living with a partner in the household was included as well as the division of childcare tasks among couples. Three categories were formed for the latter variable: “anchor person is primarily responsible”, “partner is primarily responsible” and “50% division of labor is carried out by both partners”. Single parents were assigned to the category “anchor person is primarily responsible”. 

As factors related to employment, the following variables were included: Employment status as “full-time”, “part-time”, and “self-employed”, with less than 30 h worked per week being considered “part-time”. Shift work was included as well as the self-assessed working time of the partner in the categories “full-time”, “part-time”, “self-employed”, and “not employed”. Unless otherwise specified, all variables have been included at T1.

### 2.5. Statistical Analyses

Point estimates with corresponding 95% confidence intervals (CIs) show the between-groups association of WTFC and self-reported general health at T1 stratified by gender (research question 1). 

In order to answer research question 2, a logistic regression with the four time-based values of WTFC as the independent variable was calculated. Odds ratios (OR) were estimated for self-reported general health at T1. The odds ratio indicates the factor by which the odds for health impairments among persons with high WTFC is elevated or reduced compared to persons with low WTFC at both times (reference group). Model 0 included age and area of residence as control variables. Model 1 was additionally controlled for health at T0. As for models 2, 3, and 4, the variables described in Section 2.2 were included stepwise in addition to the control variables from model 1: In model 2, the variables household income and education were added. Model 3 included family-related variables, i.e., the number of children under 18 in the household, age of the youngest child, information on the division of labor between partners for childcare, and marital status, and whether the respondent lives with a partner in the household. In model 4, work-related variables were integrated: employment status, shift work, and the employment status of partners. Model 5 included all the variables mentioned, excluding whether the respondent lives with a partner in the household, as this information was already included in the employment status variable of the partners. 

In a further step (research question 3), stratification was carried out within the groups of mothers and fathers with high and low WTFC by highest educational attainment. This was done by including the interaction of WTFC and education and estimating predictive margins. Predictive margins are the probabilities for self-reported general health predicted in the controlled model for each subgroup (mothers WTFC high/low and fathers WTFC high/low with high and medium/low education, respectively). The models for predictive margins were controlled for self-reported general health at T0, age, and area of residence.

To avoid statistical misinterpretations, no significance level was defined for the point estimators, but results were assessed by calculating the 95% CIs [43,44]. To test differences by gender, interaction terms were included in the regression models. Corresponding p-values from the Wald test are reported. All statistical analyses were performed with RStudio (version 1.2.1335, Boston, MA, USA).

## 3. Results

### 3.1. Sample Description

Table 1 describes the stratified sample for mothers and fathers according to high and low WTFC. Mothers and fathers with high WTFC reported slightly higher levels of education and a higher net household income compared to parents with low WTFC. Mothers with high WTFC also worked full-time more often, and shift work was more common among those with high WTFC.

### 3.2. Work-to-Family Conflicts and Health at T1

In all, 32.7% of working mothers and 40.7% of working fathers from the pairfam panel reported high WTFC. Figure 1 shows that 48.7% of mothers with high WTFC reported satisfying-to-bad self-reported general health, compared to significantly fewer (25.8%) mothers with low WTFC (research question 1). A significant difference was also shown within the group of fathers: 36.4% with high WTFC reported satisfying-to-bad self-reported general health in contrast to 21.2% of fathers with low WTFC. The interaction of WTFC and gender was not significant (*p* = 0.302), indicating no systematic differences regarding the association of WTFC and self-reported general health between mothers and fathers.

### 3.3. Work-to-Family Conflicts and Health over Time 

The analysis of WTFC over time at T0 and T1 is shown in Figure 2 (research question 2). Compared to persons who reported low WTFC at both times (reference group), mothers with high conflicts at T0 and T1 as well as mothers who reported high conflicts at T1 only showed significantly poorer general health. This difference was not apparent in mothers who reported high WTFC at T0 only. A similar picture emerged for fathers: Statistically, however, the difference in self-reported general health compared to the reference group was only confirmed for those fathers who reported high WTFC at both points in time. Furthermore, the association of WTFC and health did not differ between mothers and fathers, as the interaction between WTFC and gender was not significant (*p* = 0.707).

These results were confirmed in multivariable analyses. Table 2 shows that in the crude model (0) the odds of reporting satisfying-to-bad general health at T1 were more than threefold higher for mothers with high WTFC at times T0 and T1 (reference group: low WTFC at T0 and T1). When including health at T0 in the model (1), the odds were slightly reduced but remained almost 2.8 times higher compared to the reference group and significant. Mothers who reported high WTFC at T1 only also showed a more than 2.4-fold higher odds of reporting poor health in model 1. On the other hand, no significant differences in health at T1 were shown when mothers reported high WTFC at T0 only. In models 2 to 4 socioeconomic (2), family-related (3), and work-related (4) variables were added. In addition, a fully adjusted model was calculated (5). In all five models, the described associations between WTFC and health remained significant.

Table 3 shows a similar picture for fathers: Here too, significant associations between WTFC and health were shown for the groups in which high WTFC occur at T0 and T1 or at T1 only. As for mothers, the variables included in the models did not fully explain the associations between WTFC and self-reported general health.

Interactions of gender and WTFC were calculated for all five models. Wald tests showed no significant interactions (model 0: *p* = 0.669, model 1: *p* = 0.707, model 2: *p* = 0.570, model 3: *p* = 0.894, model 4: *p* = 0.8486, model 5: *p* = 0.875). This indicates that no differences between mothers and fathers in the associations of WTFC and health were present in the data.

### 3.4. The Role of Education 

With regard to the interactions between WTFC and education (research question 3), associations between WTFC and self-reported general health were similar for mothers in both educational groups (see Figure 3). 

For fathers, on the other hand, the results appeared to be somewhat different. In the group of fathers who reported high WTFC at T0 only, fathers with a high education tended to show the same level of satisfying-to-bad health as those who had never reported high WTFC.

In contrast, however, the interaction of WTFC, education, and gender was not statistically significant (*p* = 0.634), i.e., no systematic differences in the association of education and WTFC between working mothers and fathers were detected.

## 4. Discussion

Our analyses are the first to show that parents with high WTFC living in Germany report significantly poorer self-reported general health than mothers and fathers with low WTFC. This confirms results from European and U.S.-American studies in which high WTFC was associated with poorer self-reported general health [16,38]. Considering WTFC as a source of stress, the results are consistent with stress process theory [13,14], which argues that stress is a significant predictor of individual health.

From a longitudinal perspective, parents with high WTFC at T0 and T1 or at T1 only showed significantly poorer general health compared to parents who reported low WTFC at both points in time (reference group). This result partially confirms previous work by Leineweber et al. (2013), who presented significant associations for WFC and self-reported general health for women and men over time. These, however, were to a large extent explained by the health status at T0 [20]. A publication looking at the association between WTFC and health in a fixed-effects model also found a strong association of WTFC and self-reported general health [45], supporting our finding of a negative association between WTFC and health over time. Future research, however, should aim at a deeper understanding of the etiology of health burdens that stem from WTFC.

In our data, WTFC seems not to have a cumulative effect on self-reported general health. Hence, parents with high WTFC at both T0 and T1 did not report worse health than parents who reported high WTFC at one point in time only. The results also showed that the assessment of self-reported general health at T1 improved significantly when WTFC only occurred at T0. As far as we know, this is the first analysis indicating a rather short-termed effect of WTFC on health. However, further analyses should examine this result in detail and analyze more points in time. 

In our multivariate analyses, the association between WTFC and health was not explained by the inclusion of health at T0 or demographic, family, and work-related variables. This may suggest that the association of WTFC and health is independent of the social determinants included in the models, thus affecting mothers and fathers with diverse social backgrounds to a similar extent. 

The results also implicate no gender differences in direction and strength regarding the association of WTFC and health. This is in line with an internationally growing number of studies that do not show differences in health-related consequences of WTFC by gender [38]. On the other hand, this seems remarkable against the background of the different work- and family-related circumstances mothers and fathers are facing in Germany. We interpret that different role distributions and burdens of employment and family roles among mothers and fathers mean that differences may appear earlier with relation to the occurrence of WTFC than differences in health-related consequences. It is also conceivable that mothers and fathers react differently to WTFC: Mothers might, e.g., reduce their working time when WTFC arises, an aspect that could not be taken into account in the present analysis, partly due to the small number of cases. Future research should consider this aspect as well as its impact on the prevalence of WTFC. Adding another perspective, Kobayashi et al. claim that the existing instruments measuring WTFC do not equally apply to mothers and fathers, producing mixed results regarding gender differences in WTFC and health [32]. 

With regard to the interactions between WTFC and education, no significant result among mothers was found. This indicates that the effect of WTFC on health exists regardless of educational attainment. Within the group of fathers reporting high WTFC at T0 only, however, those with intermediate and lower educational levels may be more affected by health effects than fathers from the high educational group. We suspect that higher education is associated with higher financial and psychosocial resources that may buffer the health effects of WTFC. With great caution, we further derive that this buffer effect seems to be absent among mothers, suggesting that financial and psychosocial resources do not protect mothers from detrimental effects of WTFC on health. Overall, the interpretation of the predictive margins has to be carried out with care: Low case numbers and very large confidence intervals may lead to overlooking educational differences. Further research on the role of resources is urgently needed to deepen the understanding of these findings.

### Limitations

Although the analyses of the association between WTFC and health over two points in time represent a special quality of the present work, it should be critically assessed that WTFC may not be reliably measured by a two-point assessment. Thus, the grouping of persons by exposure to high and low WTFC at both points in time are theoretical to a certain extent as it remains unclear, e.g., since when the conflicts existed and whether they persist between the measurements in the survey.

In addition, the rather small number of cases of working mothers and fathers should be mentioned, which particularly affected parents with a high level of education. These led to very large confidence intervals and corresponding inaccuracy of the point estimators. Future studies can counter this by implementing measures for WTFC early on in longitudinal studies, when sample sizes are still large. 

Health was measured with a self-reported single-item question. Although this general health measure is considered to be a valid proxy for both physical and mental health [37], future research should study different health outcomes separately and in more detail. It would be interesting, for example, to distinguish whether there are greater effects of WFC on physical or mental health. Moreover, our finding on WTFC and health should be substantiated by including objective health measures, e.g., through physiological indicators. In addition, there may be an association of WTFC and health in the reverse causal direction (selection effects), where poorer general, physical, and mental health leads to more WTFC [19,20,21,22].

The present sample from waves 6 and 8 of the pairfam data set might also be a very selective group due to drop-out from the study. Therefore the prevalences, as mentioned above, are not representative for the total population of working mothers and fathers in Germany.

The present work also focused exclusively on WTFC. However, more research is needed regarding the health-related consequences of FTWC, particularly with a perspective on differences by gender. Furthermore, it should be noted that the scientific community is increasingly publishing work on the concepts of “work–family enrichment” and “work–life balance”. As a continuation of the long research tradition on multiple roles and the perspective of the “role enrichment thesis”, these show that for many working mothers and fathers the reconciliation of employment and family roles has positive effects on health [46,47,48,49].

## 5. Conclusions

The relevance of this analysis should be emphasized despite the methodological limitations. Against the background of increasing international demands for intersectional perspectives in public health, the calculation of complex interactions in quantitative research allows readers to differentiate within the groups of women and men, thus identifying particular burdens and adapting prevention measures accordingly. In addition, strategies such as the one presented also allow for identifying groups that are less burdened due to personal and social resources, leading to starting points for resource-oriented health promotion.

Future studies should therefore further investigate differences within the groups of working mothers and fathers. These can provide information on how individual social determinants, such as income and occupation, interact with gender. Furthermore, structural characteristics such as social deprivation of the area of residence should be included in order to investigate interactions of these with individual preconditions. In addition to the required differentiation of the groups, datasets should be available that allow calculations of intra-individual variation to investigate causality in the context of WTFC and health. For this purpose, longitudinal cohort studies with sufficiently large numbers of participants are required. 

In the development of measures to reduce WTFC between family and work, the results presented here can provide an initial indication that the level of education plays a role in the context of WTFC and health for fathers. Against this background, existing measures both at the political level and in occupational health management should not only be implemented and evaluated for working mothers but also for fathers. 

## Figures and Tables

**Figure 1 ijerph-17-03966-f001:**
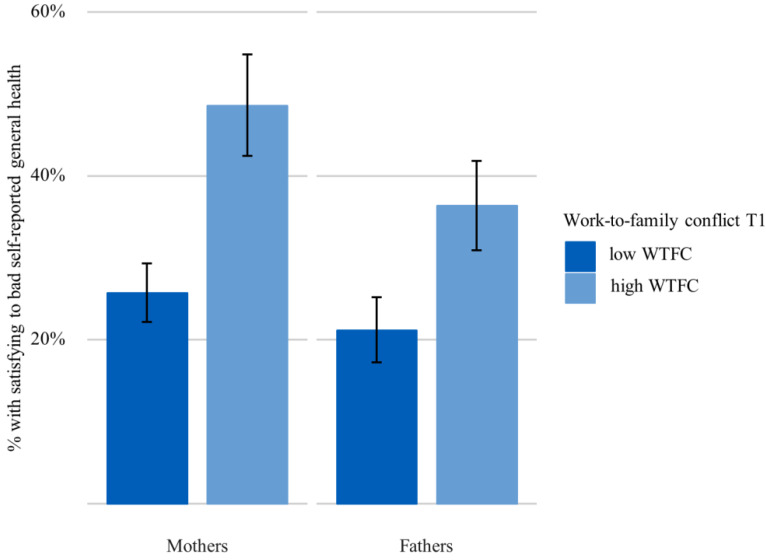
Percentage of parents with satisfying-to-bad self-reported general health (T1) by WTFC (T1), *n* = 791 mothers and *n* = 723 fathers, (prevalence in % and 95% CI (confidence interval)).

**Figure 2 ijerph-17-03966-f002:**
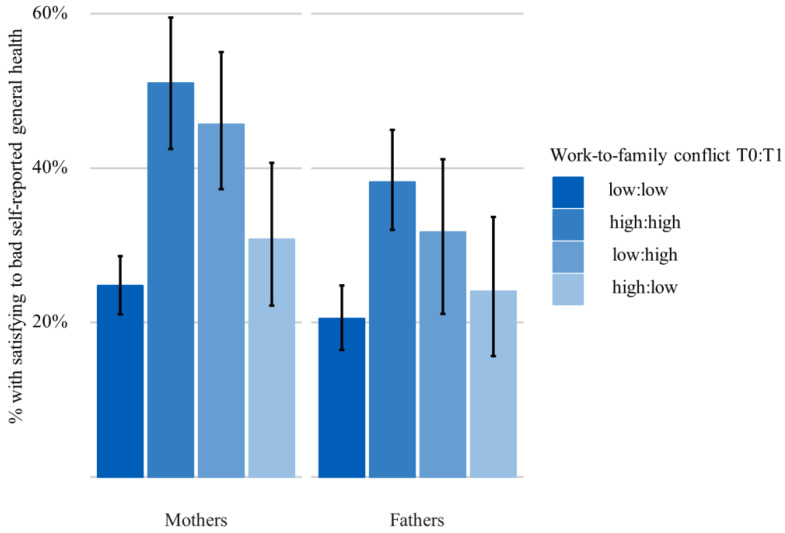
Parents with satisfying-to-bad self-reported general health (T1) by WTFC (T0 and T1), *n* = 791 mothers and *n* = 723 fathers, (prevalence in % and 95% CI).

**Figure 3 ijerph-17-03966-f003:**
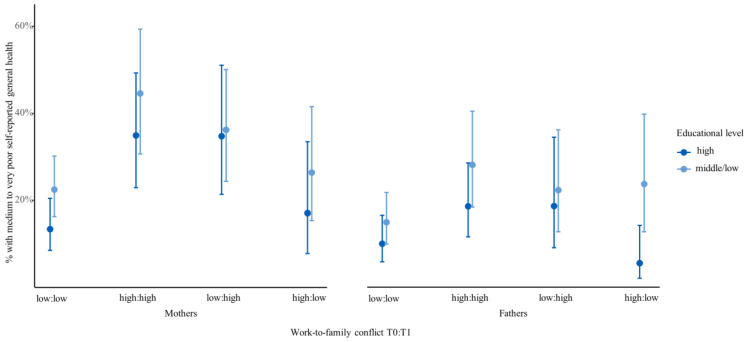
Predictive margins for parents with satisfying-to-bad self-reported general health (T1) by WTFC (T0 and T1) and educational level, adjusted for age, region, and self-reported health T0, *n* = 791 mothers and *n* = 723 fathers (in % and 95% CI).

**Table 1 ijerph-17-03966-t001:** Sample description by low and high work-to-family conflict (WTFC), unweighted frequencies, and percentages at T1.

Variables	Mothers		Fathers		Total
	Low WTFC (*n* = 532)	High WTFC (*n* = 259)	Low WTFC (*n* = 429)	High WTFC (*n* = 294)	*n* = 1514
**Self-Reported General Health**					
Good to very good	395 (74.2%)	133 (51.4%)	338 (78.8%)	187 (63.6%)	1053 (69.6%)
Satisfying to bad	137 (25.8%)	126 (48.6%)	91 (21.2%)	107 (36.4%)	461 (30.4%)
Missing	0	0	0	0	0
**Age Group**					
18–35	153 (28.8%)	73 (28.2%)	139 (32.4%)	81(27.6%)	446 (29.5%)
36–45	379 (71.2%)	186 (71.8%)	290 (67.6%)	213 (72.4%)	1068 (70.5%)
Missing	0	0	0	0	0
**Area of Residence**					
West Germany	340 (63.9%)	158 (61.0%)	263 (61.3%)	186 (63.3%)	947(62.5%)
East Germany	192 (36.1%)	101 (39.0%)	166 (38.7%)	108 (36.7%)	567 (37.5%)
Missing	0	0	0	0	0
**Net Household Income**					
High	122 (25.4%)	76 (32.6%)	118 (28.9%)	85 (31.0%)	401 (28.7%)
Middle/low	359 (74.6%)	157 (67.4%)	291 (71.1%)	189 (69.0%)	996 (71.3%)
Missing	51	26	20	20	117
**Education (Highest Vocational Degree)**					
High	211 (39.7%)	120 (46.3%)	186 (43.4%)	145 (49.3%)	662 (43.7%)
Middle/low	321 (60.3%)	139 (53.7%)	243 (56.6%)	149 (50.7%)	852 (56.3%)
Missing	0	0	0	0	0
**Number of Children Under 18 Living in Household**					
1	178 (33.5%)	101 (39.0%)	131 (30.5%)	70(23.8%)	480 (31.7%)
2	276 (51.9%)	130 (50.2%)	221 (51.5%)	160 (54.4%)	787 (52.0%)
>2	78 (14.7%)	28 (10.8%)	77 (17.9%)	64 (21.8%)	247 (16.3%)
Missing	0	0	0	0	0
**Age of the Youngest Child**					
0–4	51 (9.6%)	32 (12.4%)	149 (34.7%)	88(29.9%)	320 (21.1%)
5–13	358 (67.3%)	166 (64.1%)	231 (53.8%)	169 (57.5%)	924 (61.0%)
14–17	123 (23.1%)	61 (23.6%)	49 (11.4%)	37 (12.6%)	270 (17.8%)
Missing	0	0	0	0	0
**Marital Status**					
Never married	85 (16.0%)	43 (16.7%)	47 (11.0%)	25 (8.5%)	200 (13.2%)
Married/civil union	385 (72.4%)	182 (70.8%)	362 (84.4%)	257 (87.7%)	1186 (78.5%)
Divorced/widowed	62 (11.7%)	32 (12.5%)	20 (4.7%)	11 (3.8%)	125 (8.3%)
Missing	0	2	0	1	3
**Partner Living in Household**					
No	100 (18.8%)	41 (15.8%)	19 (4.4%)	10 (3.4%)	170 (11.2%)
Yes	432 (81.2%)	218 (84.2%)	410 (95.6%)	284 (96.6%)	1344 (88.8%)
Missing	0	0	0	0	0
**Division of Labor Between Anchor Person and Partner in Childcare**					
Anchor person	355 (68.3%)	155 (61.5%)	28 (6.6%)	16 (5.5%)	384 (25.8%)
50/50	162 (31.2%)	86 (34.1%)	174 (41.0%)	89 (30.5%)	511 (34.3%)
Partner	3 (0.6%)	11 (4.4%)	222 (52.4%)	187 (64.0%)	423 (28.4%)
Missing	12	7	5	2	26
**Employment Status**					
Full time	147 (27.6%)	113 (43.6%)	379 (88.3%)	242 (82.3%)	881 (58.2%)
Part time	331 (62.2%)	120 (46.3%)	12 (2.8%)	7 (2.4%)	470 (31.0%)
Self employed	54 (10.2%)	26 (10.0%)	38 (8.9%)	45 (15.3%)	163 (10.8%)
Missing	0	0	0	0	0
**Shiftwork**					
No shiftwork	432 (81.4%)	186 (71.8%)	336 (78.5%)	235 (79.9%)	1189 (78.6%)
Shiftwork	99 (18.6%)	73 (28.2%)	92 (21.5%)	59 (20.1%)	323 (21.4%)
Missing	1	0	1	0	2
**Employment Status of Partner**					
Not working	16 (3.2%)	13 (5.2%)	98 (23.5%)	66 (22.9%)	193 (13.3%)
Full time/self employed	396 (80.0%)	191 (76.7%)	129 (30.9%)	89 (30.9%)	805 (55.6%)
Part-time	19 (3.8%)	14 (5.6%)	179 (42.9%)	128 (44.4%)	340 (23.5%)
No partner in household	64 (12.9%)	31 (12.4%)	11 (2.6%)	5 (1.7%)	111 (7.7%)
Missing	37	10	12	6	65

*n* = 791 mothers and *n* = 723 fathers.

**Table 2 ijerph-17-03966-t002:** Results from the logistic regression (odds ratio and 95% CI) with self-reported general health at T1 as the dependent and WTFC at T0 and T1 as the independent variable for mothers.

WTFC T0:T1	Crude Model	Health at T0	Socio-Demographics	Family Characteristics	Work Characteristics	Full Model
	(0)	(1)	(2)	(3)	(4)	(5)
*n*	791	791	714	770	743	653
WTFC T0:T1						
Low:Low	Ref.	Ref.	Ref.	Ref.	Ref.	Ref.
High:High	**3.23**	**2.80**	**3.24**	**2.95**	**2.54**	**2.94**
	**(2.17, 4.80)**	**(1.86, 4.22)**	**(2.09, 5.05)**	**(1.92, 4.53)**	**(1.65, 3.92)**	**(1.81, 4.80)**
Low:High	**2.54**	**2.38**	**2.44**	**2.34**	**2.10**	**2.13**
	**(1.67, 3.87)**	**(1.54, 3.68)**	**(1.54, 3.88)**	**(1.48, 3.69)**	**(1.39, 3.44)**	**(1.29, 3.51)**
High:Low	1.38	1.25	1.28	1.28	1.01	1.03
	(0.82, 2.31)	(0.72, 2.11)	(0.72, 2.23)	(0.72, 2.21)	(0.56, 1.78)	(0.54, 1.89)
AIC	974.64	930.12	839.23	906.63	888.59	786.98

Bold = significance (1 is not included in the 95% CI), AIC = Akaike information criterion. (0): Age, region. (1): Age, region, self-reported general health T0. (2): Age, region, self-reported general health T0, net household income, education (ISCED). (3): Age, region, self-reported general health T0, number of children in household, age of the youngest child, partner living in household, marital status, division of childcare labor. (4): Age, region, self-reported general health T0, job status, shiftwork, job status of partner. (5): Age, region, self-reported general health T0, net household income, education (ISCED), number of children in household, age of the youngest child, marital status, division of childcare labor, job status, shiftwork, job status of partner.

**Table 3 ijerph-17-03966-t003:** Results from the logistic regression (odds ratio and 95% CI) with self-reported general health at T1 as the dependent and WTFC at T0 and T1 as the independent variable for fathers.

WTFC T0:T1	Crude Model	Health at T0	Socio-Demographics	Family Characteristics	Work Characteristics	Full Model
	(0)	(1)	(2)	(3)	(4)	(5)
*n*	723	723	683	715	705	661
WTFC T0:T1						
Low:Low	Ref.	Ref.	Ref.	Ref.	Ref.	Ref.
High:High	**2.39**	**2.03**	**2.15**	**2.05**	**2.14**	**2.21**
	**(1.63, 3.51)**	**(1.36, 3.02)**	**(1.41, 3.27)**	**(1.37, 3.09)**	**(1.42, 3.23)**	**(1.42, 3.45)**
Low:High	**1.81**	**1.78**	**1.84**	**1.91**	**1.84**	**1.96**
	**(1.07, 3.07)**	**(1.03, 3.04)**	**(1.04, 3.19)**	**(1.09, 3.29)**	**(1.05, 3.17)**	**(1.09 3.47)**
High:Low	1.22	1.01	1.11	1.03	1.04	1.11
	(0.69, 2.15)	(0.55, 1.79)	(0.59, 2.02)	(0.55, 1.84)	(0.55, 1.88)	(0.57, 2.07)
AIC	836.30	803.81	749.90	810.51	791.94	746.98

Bold = significance (1 is not included in the 95% CI), AIC = Akaike information criterion. (0): Age, region. (1): Age, region, self-reported general health T0. (2): Age, region, self-reported general health T0, net household income, education (ISCED). (3): Age, region, self-reported general health T0, number of children in household, age of the youngest child, partner living in household, marital status, division of childcare labor. (4): Age, region, self-reported general health T0, job status, shiftwork, job status of partner. (5): Age group, region, self-reported general health T0, net household income, education (ISCED), number of children in household, age of the youngest child, marital status, division of childcare labor, job status, shiftwork, job status of partner.

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
