# Peer review of "Longitudinal Analysis of Work-to-Family Conflict and Self-Reported General Health among Working Parents in Germany"

_ijerph, 2020, doi:10.3390/ijerph17113966_

Round 1

Reviewer 1 Report

The paper entitled “Longitudinal analysis of work-family conflicts and self-reported general health among working parents in Germany” has a motivating introduction, offers interesting results and has the great strength of being a longitudinal study. However, some aspects need to be reviewed and improved.

Title: "work-family conflicts" should be replaced by "work-to-family conflict". The work only evaluated this type of conflict.

Methodology:
- The dichotomization of the dependent variable and the way it was done must be justified. Firstly, dichotomization can imply loss of information and clearly "bad" is not the same as "satisfactory".
- Do the items used to measure work-to-family conflict belong on a scale? If so, the reference should be added. Regardless of the above, the internal consistency of the 4 items together must be reported.
- The division of the groups according to the level of work-to-family conflict over time must be explained and justified.

Results
- You cannot answer research question 1 only with descriptive analysis. It must be supported with statistical analysis. Furthermore, the second part of this research question, referring to the difference between fathers and mothers, was not answered.

Discussion
Authors should improve this section. The interpretation of the results must be deepened. In addition, practical implications derived from the results must be added. Although the authors suggest future lines of research regarding studying other concepts (“work-family enrichment” and “work-life balance”), they do not suggest future research derived from the results of the study or the limitations of the study. For example, self-reported general health was measured, but it would be interesting to distinguish whether there are greater effects on physical or mental health. Another limitation that could lead to new studies is the lack of knowledge of the origin of the diseases associated with better or worse self-reported health. Some may be associated with work-family conflict, but many others may not.

Author Response

The paper entitled “Longitudinal analysis of work-family conflicts and self-reported general health among working parents in Germany” has a motivating introduction, offers interesting results and has the great strength of being a longitudinal study. However, some aspects need to be reviewed and improved.

Thank you for the helpful review!

Title

(1) "work-family conflicts" should be replaced by "work-to-family conflict". The work only evaluated this type of conflict.

The title and terms have been revised accordingly throughout the entire manuscript.

Methodology

(2) The dichotomization of the dependent variable and the way it was done must be justified. Firstly, dichotomization can imply loss of information and clearly "bad" is not the same as "satisfactory".

Thank you for this important comment. In public health research, it is common to dichotomize self-reported general health as the dependent variable, please see e.g. Borgmann, et al. (2019), Kobayashi et al. (2017), Leineweber, et al. (2012), Lunau et al. (2014), and Winter, et al. (2006).Thus, we also chose dichotomization to make our work comparable to prior research. Also, the resulting measures in the regression models (odds ratios) and predictive margins are in our view easy to interpret, even for less epidemiologically trained readers.

Although this approach can clearly be criticized due to the loss of information, the application of the full scale is also accompanied by controversial discussions and it remains unclear if the scale is equally spaced, i.e. whether it has to be handled as an ordinal or metric variable (Manor, et al. (2000)).

Although we agree that “bad” and “satisfactory” should only be summed in one group with great caution, Finnäs, et al. (2008) argue that self-reported general health has to be viewed as a continuum that ranges from poor to good. They conclude that categorizing the ordinal outcome into a binary response consequently does not have an impact on the estimated effects of covariates. We have included this justification in the manuscript, please see lines 170ff. 

References

Borgmann LS, Kroll LE, Müters S, Rattay P, Lampert T (2019): Work-Family Conflict, Self-Reported General Health and Work-Family Reconciliation Policies in Europe: Results from the European Working Conditions Survey 2015. SSM – Population Health, 9.

Finnäs F, Nyqvist F, Saarela J (2008): Some Methodological Remarks on Self-Rated Health, The Open Public Health Journal, 1, 32-39.

Kobayashi T, Honjo K, Eshak ES, Iso H, Sawada N and Tsugane S (2017): Work-family conflict and self-rated health among Japanese workers: How household income modifies associations. PLoS ONE 12(2), e0169903.

Leineweber C, Maltzer M, Magnusson Hanson LL and Westerlund H (2012): Work–family Conflict and Health in Swedish Working Women and Men: A 2-year Prospective Analysis (the SLOSH study). European Journal of Public Health, 23(4), 710-716.

Lunau T, Bambra C, Eikemo TA, van Der Wel KA, Dragano N (2014): A balancing act? Work-life balance, health and well-being in European welfare states. European Journal of Public Health, 24(3), 422-427.

Manor O, Matthews S, Power C (2000): Dichotomous or Categorical Response? Analysing Self-Rated Health and Lifetime Social Class. International Journal of Epidemiology, 29, 149-157).

Winter T, Roos E, Rahkonen O, Martikainen P and Lahelma E (2006): Work–Family Conflicts and Self-Rated Health Among Middle-Aged Municipal Employees in Finland. International Journal of Behavioral Medicine, 13(4), 276-285.

(3) Do the items used to measure work-to-family conflict belong on a scale? If so, the reference should be added.

A reference has been added. Please see lines 178f.

(4) Regardless of the above, the internal consistency of the 4 items together must be reported.

The Cronbach`s Alpha has been added, please see lines 190f.

(5) The division of the groups according to the level of work-to-family conflict over time must be explained and justified.

We have elaborated on the division of participants into the four groups. Please see lines 202ff.

The cut point of 10/11 points in the sum index has been chosen as it guarantees that participants have at least scored a 3 or higher on the Likert scale. We have added this consideration to the manuscript, please see lines 193f. Also, dichotomization of the work-family conflict scale is common in prior research; please see Borgmann, et al. (2019), Kobayashi, et al. (2018), Leineweber, et al. (2012), Lunau at al. (2014).

References

Borgmann LS, Kroll LE, Müters S, Rattay P, Lampert T (2019): Work-Family Conflict, Self-Reported General Health and Work-Family Reconciliation Policies in Europe: Results from the European Working Conditions Survey 2015. SSM – Population Health, 9.

Kobayashi T, Honjo K, Eshak ES, Iso H, Sawada N and Tsugane S (2017): Work-family conflict and self-rated health among Japanese workers: How household income modifies associations. PLoS ONE 12(2), e0169903.

Leineweber C, Maltzer M, Magnusson Hanson LL and Westerlund H (2012): Work–family Conflict and Health in Swedish Working Women and Men: A 2-year Prospective Analysis (the SLOSH study). European Journal of Public Health, 23(4), 710-716.

Lunau T, Bambra C, Eikemo TA, van Der Wel KA, Dragano N (2014): A balancing act? Work-life balance, health and well-being in European welfare states. European Journal of Public Health, 24(3), 422-427.

Results

(6) You cannot answer research question 1 only with descriptive analysis. It must be supported with statistical analysis.

(7) Furthermore, the second part of this research question, referring to the difference between fathers and mothers, was not answered.

The confidence intervals in Fig. 1 indicate that statistical analyses have been performed. To avoid statistical misinterpretations no significance level was defined but all results were assessed by calculating the 95% confidence intervals of the point estimators, as suggested by Greenland and Poole (2011) and Greenland, et al. (2016). However, a p-value has been calculated for the interaction of work-to-family conflicts and gender to answer the second part of research question 1, please see line 281ff.

References

Greenland S, Poole C (2011): Problems in Common Interpretations of Statistics in Scientific Articles, Expert Reports, and Testimony. Jurimetrics, 51, 113-29.

Greenland S, Senn SJ, Rothman KJ, et al. (2016): Statistical Tests, P Values, Confidence Intervals, and Power: A Guide to Misinterpretations. European journal of epidemiology, 31, 337-50.

Discussion

(8) Authors should improve this section. The interpretation of the results must be deepened.

We have complemented the discussion. Please see lines 388ff., 402f., and 426ff.

(9) In addition, practical implications derived from the results must be added. Although the authors suggest future lines of research regarding studying other concepts (“work-family enrichment” and “work-life balance”), they do not suggest future research derived from the results of the study or the limitations of the study. For example, self-reported general health was measured, but it would be interesting to distinguish whether there are greater effects on physical or mental health. Another limitation that could lead to new studies is the lack of knowledge of the origin of the diseases associated with better or worse self-reported health. Some may be associated with work-family conflict, but many others may not.

Thank you for this helpful comment. We have complemented the limitations section accordingly, please see lines 445ff.

Thank you very much and best regards

Reviewer 2 Report

Dear authors.

Thank you very much for the opportunity of reviewing this interesting paper. 

Introduction places the reader very well.

Method and results section are very well described too. However, I guess why you havent used any of the already validated questionnaires such as Work-Family Conflict scale.

I think your research would gain a lot of scientifical sound with that measure. 

Please, check the references section and adapt them to MDPI format.

Good paper. Congratulations.

Author Response

Dear authors,

thank you very much for the opportunity of reviewing this interesting paper.

Thank you for the considerate and helpful review!

(1) Introduction places the reader very well.

Thank you!

(2) Method and results section are very well described too. However, I guess why you haven’t used any of the already validated questionnaires such as Work-Family Conflict scale. I think your research would gain a lot of scientifically sound with that measure. Sorry for the confusion and thank you for the advice. The data is indeed based on a tested and validated scale. The respective reference has been added. Please see lines 178f.

(3) Please, check the references section and adapt them to MDPI format.

The references have been updated according to IJERPH reference style.

Good paper. Congratulations.

Thank you!

Thank you very much and best regards

Reviewer 3 Report

This study takes a longitudinal approach to the examination of work-family conflict in relation to health. An important contribution seems to be the reporting of results for working mothers and fathers separately. However, the methodology and especially statistical analyses are problematic (see #5). I elaborate on this below in my list of comments.

  1. Line 44: it is stated that family-to-work conflicts occur when family demands make it more difficult to pursue a career, but this is a very uncommon, narrow and long-term definition of the construct. There is a stream of literature that investigates family-to-work conflict on a daily basis, focusing on everyday interferences with daily work.
  2. The paragraph starting at line 72 is incoherent. It is about longitudinal research, gender differences, reversed causality, cultural differences, and methods in WF research. What is the message that the authors are trying to bring across?
  3. Please explain and justify why wave 6 and 8 are chosen and not two other waves or more than two waves.
  4. Line 165: what does it mean that family-to-work conflicts did not show enough variation in the respective waves? It is a pity this variable is not included.
  5. My major concern is that all variables are turned into binary variables while the key variables were measured on a five-point scale. The authors mention it as a limitation in line 387, but this is confusing because they can easily do otherwise. There are three disadvantages to this approach. First, the richness of the data gets wasted. Second, results are difficult to compare with studies that operationalized the constructs differently. Third, the output from logistic regression (needed for a categorical dependent variable) is far from straightforward and easy to interpret for readers.
  6. Do the items stem from existing and validated scales? If not, it is important to do a confirmatory factor analysis (CFA). Cronbach’s alphas should also be reported.
  7. Controlling for health at T0 actually implies that one is predicting the change in health.
  8. The four categories for WFC are potentially interesting. But what does it mean theoretically to be in one group and not in the other? Why it is important to compare four groups rather than individuals who have different scores along a 1-5 scale? The discussion touches upon cumulative effects, but this should be explained better and much earlier in the paper.
  9. Related to this, why is a longitudinal approach needed? It is important to elaborate on this. The finding that WFC (only) at T1 is related to health at T1 is hardly longitudinal.
  10. Please not that odds ratios do not show that thechance (e.g. line 274) is higher for one group but the odds.
  11. Some results are missing. I suggest reporting p-values in addition to CIs. Figures do not show which differences are significant. And I don’t understand why the (two-way and three-way) interaction results are not shown because this was clearly a research question/goal.
  12. A limitation that should probably be mentioned in 4.1 is that health is self-reported.

Author Response

This study takes a longitudinal approach to the examination of work-family conflict in relation to health. An important contribution seems to be the reporting of results for working mothers and fathers separately. However, the methodology and especially statistical analyses are problematic (see #5). I elaborate on this below in my list of comments.

Thank you for the helpful review!

(1) Line 44: it is stated that family-to-work conflicts occur when family demands make it more difficult to pursue a career, but this is a very uncommon, narrow and long-term definition of the construct. There is a stream of literature that investigates family-to-work conflict on a daily basis, focusing on everyday interferences with daily work.

The definition has been revised accordingly. Please see line 43f.

(2) The paragraph starting at line 72 is incoherent. It is about longitudinal research, gender differences, reversed causality, cultural differences, and methods in WF research. What is the message that the authors are trying to bring across?

The paragraph has been revised accordingly. Please see lines 77ff..

(3) Please explain and justify why wave 6 and 8 are chosen and not two other waves or more than two waves.

The work–family conflict scale was only included in every second wave of the pairfam data collection, starting only at wave 6. By the time of data analysis wave 9 was the latest wave released; resulting in only two waves where the work–family conflict scale was included. We tried to clarify this in the manuscript, please see line 158.

(4) Line 165: what does it mean that family-to-work conflicts did not show enough variation in the respective waves? It is a pity this variable is not included.

The lack of variation means that only about 10%of men and women reported high family-to-work conflicts across wave 6 and 8, compared to more than 32% (mothers) and 40% (fathers) for work-to-family conflicts. Thus, the group with high family-to-work conflicts would be too small to run subgroup analysis. We have added this explanation to the manuscript, please see lines 181f.

(5) My major concern is that all variables are turned into binary variables while the key variables were measured on a five-point scale. The authors mention it as a limitation in line 387, but this is confusing because they can easily do otherwise. There are three disadvantages to this approach. First, the richness of the data gets wasted. Second, results are difficult to compare with studies that operationalized the constructs differently. Third, the output from logistic regression (needed for a categorical dependent variable) is far from straightforward and easy to interpret for readers.

Thank you for this important comment. In public health research, it is common to dichotomize self-reported general health as the dependent variable. Although this approach can be criticized due to the loss of information, the use of the full scale is also accompanied by controversial discussions as it remains unclear if the scale is equally spaced, i.e. whether it has to be handled as an ordinal or metric variable (Manor et al. (2000)).

In addition, Finnäs et al. (2008) suggest that self-reported general health has to be viewed as a continuum that goes from poor to good. They conclude that categorizing the ordinal outcome into a binary response consequently does not have an impact on the estimated effects of covariates. We opted for dichotomization as in our view the resulting measures in the regression models (odds ratios) and predictive margins are easy to interpret, even for less epidemiologically trained readers. We have included this justification in the manuscript, please see lines 170f.

Also, comparability was one of the reasons we chose dichotomization, as dichotomization is common in the literature of work-family conflicts and health; please see e.g. Borgmann, et al. (2019), Kobayashi et al. (2017), Leineweber, et al. (2012), and Winter, et al. (2006). In general, Finnäs, et al. (2008) as well as Manor et al. (2000) emphasize the importance of dichotomization of self-reported general health in past and present public health literature.

Regarding the dichotomization of work-to-family conflicts, the cut point of 10/11 points in the sum index has been chosen as it guarantees that participants have answered at least three items with a 3 or higher on the Likert scale. We have added this consideration to the manuscript, please see lines 193f. Dichotomization of the work-family conflict scale is common in prior research to make results clear and easy to interpret; please see Borgmann, et al. (2019), Kobayashi, et al. (2018), Leineweber, et al. (2012), and Lunau at al. (2014).

References

Borgmann LS, Kroll LE, Müters S, Rattay P, Lampert T (2019): Work-Family Conflict, Self-Reported General Health and Work-Family Reconciliation Policies in Europe: Results from the European Working Conditions Survey 2015. SSM – Population Health, 9.

Finnäs F, Nyqvist F, Saarela J (2008): Some Methodological Remarks on Self-Rated Health, The Open Public Health Journal, 1, 32-39.

Kobayashi T, Honjo K, Eshak ES, Iso H, Sawada N and Tsugane S (2017): Work-family conflict and self-rated health among Japanese workers: How household income modifies associations. PLoS ONE 12(2), e0169903.

Leineweber C, Maltzer M, Magnusson Hanson LL and Westerlund H (2012): Work–family Conflict and Health in Swedish Working Women and Men: A 2-year Prospective Analysis (the SLOSH study). European Journal of Public Health, 23(4), 710-716.

Lunau T, Bambra C, Eikemo TA, van Der Wel KA, Dragano N (2014): A Balancing Act? Work-Life Balance, Health and Well-Being in European Welfare States. European Journal of Public Health, 24(3), 422-427.

Manor O, Matthews S, Power C (2000): Dichotomous or Categorical Response? Analysing Self-Rated Health and Lifetime Social Class. International Journal of Epidemiology, 29, 149-157).

Winter T, Roos E, Rahkonen O, Martikainen P and Lahelma E (2006): Work–Family Conflicts and Self-Rated Health Among Middle-Aged Municipal Employees in Finland. International Journal of Behavioral Medicine, 13(4), 276-285.

(6) Do the items stem from existing and validated scales? If not, it is important to do a confirmatory factor analysis (CFA). Cronbach’s alphas should also be reported.

A reference has been added. Please see lines 178f. We also included the Cronbach`s Alpha, please see lines 190f.

(7) Controlling for health at T0 actually implies that one is predicting the change in health.

We added a crude model without self-reported general health (model  0) to the results. Please see pages 10 and 11 in the manuscript.

(8) The four categories for WFC are potentially interesting. But what does it mean theoretically to be in one group and not in the other? Why it is important to compare four groups rather than individuals who have different scores along a 1-5 scale? The discussion touches upon cumulative effects, but this should be explained better and much earlier in the paper.

We have created the four groups in order to examine potential cumulative effects of work-family conflict in health, as this has not been looked at in prior research. We have elaborated on the division of participants into the four groups. Please see lines 81ff., 129f., and 202ff. In our view, dichotomization leads to clearer results that are easy to interpret and can be displayed graphically in more understandable ways.

(9) Related to this, why is a longitudinal approach needed? It is important to elaborate on this. The finding that WFC (only) at T1 is related to health at T1 is hardly longitudinal.

The longitudinal approach was chosen in order to examine cumulative effects of work-family conflicts on health. First, we did not expect the absence of a longitudinal association between work-family conflicts and health prior to our research. Second, in our view the results even endorse longitudinal research, as it seems crucial for a better understanding of the dynamics work-family conflicts and health develop over time.

(10) Please note that odds ratios do not show that the chance (e.g. line 274) is higher for one group but the odds.

This stems from an error caused by translation. We have changed the term throughout the paper.

(11) Some results are missing. I suggest reporting p-values in addition to CIs. Figures do not show which differences are significant. And I don’t understand why the (two-way and three-way) interaction results are not shown because this was clearly a research question/goal.

Thank you for this important comment. To avoid statistical misinterpretations no significance level was defined but all results were assessed by calculating the 95% confidence intervals of the point estimators, as suggested by Greenland and Poole (2011) and Greenland, et al. (2016). However, p-values have been added for all interaction analyses, please see lines 281ff., 296ff., 351ff., and 369ff.

References

Greenland S, Poole C (2011): Problems in Common Interpretations of Statistics in Scientific Articles, Expert Reports, and Testimony. Jurimetrics, 51, 113-29.

Greenland S, Senn SJ, Rothman KJ, et al. (2016): Statistical Tests, P Values, Confidence Intervals, and Power: A Guide to Misinterpretations. European journal of epidemiology, 31, 337-50.

(12) A limitation that should probably be mentioned in 4.1 is that health is self-reported.

We have complemented the limitations section accordingly, please see lines 443ff..

Thank you very much and best regards

Round 2

Reviewer 1 Report

The authors improved the manuscript following my suggestions. 

Author Response

Thank you very much for the helpful review! As far as I understand no open task or comment from this reviewer remained in round 2. Is that correct?

Thank you very much and best regards

Lea-Sophie Borgmann

Reviewer 3 Report

Thank you for your responses, in particular clarifying the common practices in public health research when it comes to dichotomizing constructs and reporting results. I should say that I am an organizational psychologist and therefore used to slightly different practices. Your changes to the manuscript have certainly improved its quality and accessibility. Below are just a few more comments that you might want to consider.

  • In the added sentences on cumulative effects (71f) and also later (173), you refer to the frequency of WFC experiences. I am not sure whether this is correct, as the measurement seems to be about the level of WFC.
  • Discussion of the limitation of the health measure now centers on it being a proxy and the need to distinguish between different types of health. What I would add is that this is a self-reported measure rather than an objective measure of health, e.g. through physiological indicators.
  • I now understand better why you focused on the four groups, but I still feel it should be explained in less technical terms what it means theoretically to be in one group and not in the other.

Author Response

Thank you for your responses, in particular clarifying the common practices in public health research when it comes to dichotomizing constructs and reporting results. I should say that I am an organizational psychologist and therefore used to slightly different practices. Your changes to the manuscript have certainly improved its quality and accessibility. Below are just a few more comments that you might want to consider.

Thank you, again, for the helpful review!

(1) In the added sentences on cumulative effects (71f) and also later (173), you refer to the frequency of WFC experiences. I am not sure whether this is correct, as the measurement seems to be about the level of WFC.

We have revised both sentences accordingly. Please see lines 71 and 178.

(2) Discussion of the limitation of the health measure now centers on it being a proxy and the need to distinguish between different types of health. What I would add is that this is a self-reported measure rather than an objective measure of health, e.g. through physiological indicators.

We added a sentence on the self-reported nature of our health measure. Please see lines 392ff.

(3) I now understand better why you focused on the four groups, but I still feel it should be explained in less technical terms what it means theoretically to be in one group and not in the other.

We tried to address your comment thoroughly by adding theoretical considerations in lines 74ff., 172ff., and 380ff. If we should clarify further, please give an example of what you expected alternatively.

Thank you very much and best regards